# Accuracy of the Lotka-Volterra model fails in strongly coupled microbial consumer-resource systems

Michael P. Mustri[1,2]*, Quqiming Duan[1], Samraat Pawar[1]*

1 Department of Life Sciences, Imperial College London, Ascot, United Kingdom, 2 Department of Ecology and Evolutionary Biology, University of Arizona, Tucson, Arizona, United States of America

* mustrim@arizona.edu (MPM); s.pawar@imperial.ac.uk (SP)

## Abstract

The generalized Lotka-Volterra (GLV) model is a cornerstone of theoretical ecology for modeling the dynamics emerging from species interactions within complex ecological communities. The GLV is also increasingly being used to infer species interactions and predict dynamics from empirical data on microbial communities, in particular. However, despite its widespread use, the accuracy of the GLV's pairwise interaction structure in capturing the unseen dynamics of microbial consumer-resource interactions—arising from resource competition and metabolite exchanges—remains unclear. Here, we rigorously quantify how well the GLV can represent the dynamics of a general mathematical model that encapsulates key consumer-resource processes in microbial communities. We find that the GLV significantly misrepresents the feasibility, stability, and reactivity of microbial communities above a threshold biologically feasible level of consumer-resource coupling, because it omits higher-order nonlinear interactions. We show that the probability of the GLV making inaccurate predictions can be quantified by a simple, empirically accessible measure of timescale separation between consumers and resources. These insights advance our understanding of the temporal dynamics of resource-mediated microbial interactions and provide a method for gauging the GLV's reliability across various empirical and theoretical scenarios.

## Author summary

Microbial communities interact by competing for shared nutrients and by cross-feeding on each other's metabolic byproducts. A widely used approach to model these dynamics is the generalized Lotka-Volterra (GLV) model, which considers only pairwise interactions, under the assumption that resources adjust much faster than populations. We tested when this simplification holds by thoroughly

**Data availability statement:** All relevant code and data are contained in the following GitHub Repository: https://github.com/MustriM95/Thesis.

**Funding:** The author(s) received no specific funding for this work.

**Competing interests:** The authors have declared that no competing interests exist.

comparing a general resource-explicit consumer-resource model to its GLV approximation. We find that across a wide range of biologically realistic scenarios, the GLV mis-predicts which species persist, how stable the community is, and whether disturbances are amplified before settling (reactivity). These errors are amplified when cross-feeding is strong and when many species prefer similar resources (high niche overlap). The GLV's pairwise interaction framework breaks down in such strongly coupled, cross-feeding microbial systems because resource and consumer timescales are not well separated. In such cases, tracking key resources and using hybrid modeling will yield more reliable predictions.

## Introduction

Microbial communities play an integral role in ecosystems across scales, ranging from their influence on the metabolism of individual organisms [1], to whole-ecosystem biogeochemical cycling [2]. The functioning of any microbial community emerges from the interactions among its constituent species or "strains". Developing methods to quantify and predict the underlying mechanisms and dynamics of these interactions is a major challenge in both theoretical and applied ecology [1,3–8]. The rapid development of molecular and 'omics technologies means that relatively high-frequency and taxonomically well-resolved data on microbial communities are burgeoning [9–11]. These data provide information on the metabolic capacities ("traits") of individual microbial cells and the interactions between them, offering a means of linking changes in the composition of genetic and functional traits in the community to its emergent phenotype dynamics [12–14].

Real-world microbial communities are typically composed of complex interaction networks between trillions of cells, thousands of strains, and hundreds of species. This complexity poses a significant challenge to developing a general theory for microbial communities. To tackle this challenge, both inferential and mechanistic mathematical models of microbial communities are needed [15–18]. Inferential models are designed to deduce interactions and their effect on the dynamics of species abundances from empirical time-series data, remaining agnostic to the mechanisms that shape those interactions [19]. That is, these studies infer direct pairwise interactions between species from time series data obtained from various empirical approaches, ranging from co-culture experiments [20] to co-occurrence network snapshots reconstructed from gene sequencing, including metagenome assemblies [21]. This simplicity is advantageous in predicting the dynamical outcome of large microbial communities, where tracking every interaction and resource is impractical, and understanding the underlying mechanisms is not the primary focus.

In contrast to inferential models, mechanistic models aim to predict community dynamics from first principles by incorporating information on metabolic and physiological traits, alongside cellular processes that drive the growth of populations through species-species and species-environment interactions [4,17,22,23].

Inferential and mechanistic approaches complement each other and should ideally agree when compared at relevant spatial and temporal scales for any given microbial community. However, whether inferential pairwise models can accurately capture the inherent complexity of microbial interactions—particularly when these are indirectly generated (e.g., through cross-feeding and higher-order interactions)—and the resulting dynamics remains uncertain [16,18,24–28].

Currently, the most widely used inferential model is the generalized Lotka-Volterra system of ODEs (henceforth, the "GLVM") [7,17,23,29]. In the GLVM, each population has an intrinsic growth rate, and any pair interacts through two coefficients that quantify their reciprocal effects on each other's growth and abundance. Advances in high-throughput sequencing have enabled the collection of time-series data on microbial abundances. The GLV model can be fitted to these data to estimate interaction coefficients, providing insights into the structure and dynamics of the community.

Beyond empirical applications, the GLVM has also been used in a large body of theoretical research to derive dynamical properties of microbial communities, including feasibility (existence of a non-trivial equilibrium) and stability to perturbations as well as addition or removal of species [30–33]. However, the validity of these theoretical results also depends on how well the pairwise model approximates the underlying, mechanistic consumer-resource dynamics as an "effective" representation [27,34]. In particular, it remains unclear whether the pivotal assumption of time-scale separation between consumer and resource populations holds under a sufficiently wide range of scenarios [27,34].

Here we compare the dynamic behavior of a relatively general and widely used mechanistic model of microbial consumer-resource dynamics (MiCRM) [4,34,35] and its effective Generalized Lotka-Volterra approximation (GLVA, mathematically the same as the GLVM) and quantify the concordance between them across a range of biologically meaningful scenarios. Specifically, we focus on the GLVA's error rate, which is the frequency and magnitude of deviations in population dynamics from the approximated model compared to the "true" consumer-resource dynamics, as well as the local stability and reactivity of the equilibria.

Previous research has investigated the behavior of statistically fitted GLV models relative to the corresponding dynamics of substrate mediated interactions finding, for example, that additivity of interactions (a necessary presupposition to the GLV) is not always satisfied [16,18,25]. However, using inferred GLV interactions to validate the effectiveness of pairwise modeling suffers from two main drawbacks, namely: identifiability and the difficulty of accounting for nonlinearities. The identifiability issue stems from the fact that, for any given community, there are an infinite number of GLV parameter combinations which can reproduce the same equilibrium and similar dynamics. Hence, inferred parameters will vary depending on the fitting algorithm and the data. On the other hand, the presence of nonlineairties in the underlying dynamics present a particular challenge because they can be an additional source of error, but the extent to which they influence model inference is hard to quantify precisely [36].

Assessing the performance of an exact first-order representation of resource-mediated interactions (that is, the GLVA) can provide more insight into the specific features which make the system more or less amenable to approximation by pairwise interaction. In particular, we explore how the relative values of different parameters, such as leakage, limit how well the GLVA can capture the dynamics in principle.

We identify parameter spaces of the MiCRM that result in greater disagreement between the models. With the goal of providing both empiricists and theoreticians with criteria for identifying when substrate mediated interactions can be well approximated by pairwise interactions, we also provide a method for quantifying the degree of time-scale separation, which is at the root of the GLVA's failure.

## Materials and methods

Our overall approach is to first derive the GLVA from the MiCRM for each particular parameterization of the MiCRM, and then numerically integrate each MiCRM-GLVA ODE system paired to compare their dynamical behaviors.

## The microbial consumer resource model

The Microbial Consumer Resource Model [4,35] is given by (Table 1):

$$\frac{1}{C_i}\frac{dC_i}{dt} = \sum_{\alpha=1}^{M}(1 - l_{\alpha}^{i})u_{i\alpha}R_{\alpha} - m_i \tag{1}$$

$$\frac{dR_{\beta}}{dt} = \rho_{\beta} - \sum_{i=1}^{N}u_{i\beta}R_{\beta}C_i + \sum_{\alpha=1}^{M}\sum_{i=1}^{N}l_{\alpha\beta}^{i}u_{i\alpha}C_iR_{\alpha} \tag{2}$$

Here, Eqns 1 & 2 describe the time evolution of population (biomass) abundances of the $i$th consumer $C_i$ and $\alpha$th resource $R_{\alpha}$, respectively. For consumers, the growth rate is simply the sum of all resources consumed ($u_{i\alpha}R_{\beta}$), offset by the proportion of resources leaked ($1 - l_{\alpha}^{i}$, where $l_{\alpha}^{i} = \sum_{\beta}l_{\alpha\beta}^{i}$) and a linear loss term representing respiration ($m_i$). Resources are described by an arbitrary supply function ($\rho_{\beta}$), the amount of resources taken up by all consumers ($u_{i\alpha}R_{\alpha}C_i$), and the proportion of resources $R_{\alpha}$ leaked back into the system by consumers ($l_{\alpha\beta}^{i}u_{i\beta}C_iR_{\alpha}$).

Since the MiCRM is designed to depict two coupled interaction types: competition for shared resources and facilitation through metabolic cross-feeding, we expect the resulting system's behavior to scale non-additively with respect to the number, magnitude, and type of interactions. This non-additivity stems from the dependence of interactions on resource abundances which, depending on community structure, may not be time-invariant. Therefore, given sufficient coupling (formally defined in Section C in S1 Appendix) between the consumer and resource biomass pools, the dynamical properties of the community will be a nonlinear function of its size (number of consumers and resources) [37].

## The generalised Lotka-Volterra approximation

The derivation of the GLVA is based on the approach originally used by MacArthur [34,38]; first, assume resources reach equilibrium much faster than consumers, and then perform a first-order Taylor expansion around consumer equilibrium abundances. This yields a resource-independent approximation to the MiCRM, which can be rearranged to calculate population-level growth rates and (inter-population) pairwise interaction rates (Section A in S1 Appendix). Fig 1 provides an example of how the GLVA dynamics compare to those of the underlying MiCRM.

From the outset, the GLVA makes two strong assumptions: that interactions between populations are additive: the overall impact on a strain's population in the presence of other strains is the sum of each of its pairwise interactions, and that the time scale of resource dynamics can be decoupled from those of consumers [16,39]. As we show below, and as many others have noted beforehand [16,27,31], neither can be guaranteed in complex microbial communities for a range of biologically realistic scenarios.

## Model parameterisation and simulations

To simplify the numerical simulations, maintenance biomass ($m_i$) and resource supply ($\rho_{\alpha}$) were held equal and constant among all consumers and resources($m_i = \rho_{\alpha} = 0.2$). This choice of parameters assumes that consumers are energetically equivalent and comes with no loss of generality, since neither $m_i$ nor $\rho_i$ change the qualitative dynamics.

**Table 1**. The MiCRM's parameters.

| Symbol | Definition | Units |
|---|---|---|
| $C_i$ | Biomass of $i^{th}$ consumer | mass |
| $R_{\alpha}$ | Abundance of the $\alpha^{th}$ resource | mass |
| $u_{i\alpha}$ | Uptake rate of species $i$ per unit of resource $\alpha$ | mass$^{-1}$ time$^{-1}$ |
| $m_i$ | Maintenance biomass utilization of $i^{th}$ species | time$^{-1}$ |
| $\rho_{\alpha}$ | Supply of resource $R_{\alpha}$ | mass time$^{-1}$ |
| $l_{\alpha\beta}^{i}$ | Resource $\beta$ leaked by species $i$ through the consumption of $R_{\alpha}$ | unitless |

PLOS Computational Biology

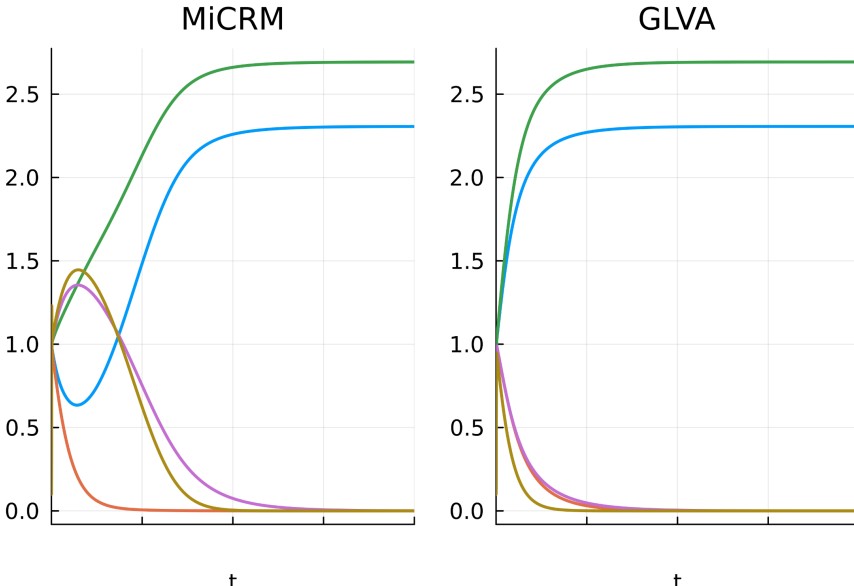

**Fig 1**. **Example dynamics of the MiCRM and its corresponding GLVA.** Plotted for a specific combination of parameters, demonstrating the differences in their transient dynamics.

The consumer uptake matrix was drawn from a Dirichlet distribution defined by the consumer preference matrix $\theta_{i\alpha}$ and a consumer specificity parameter $\Omega_i$. $\theta_{i\alpha}$ represents the probability that consumer $i$ will uptake resource $\alpha$ at a higher rate relative to other resources, while $\Omega_i$ defines how evenly the uptake is spread across the space of possible resources, with large values of $\Omega_i$ yielding generalist species and smaller ones yielding specialists.

$$\{u_{i\alpha}\} \sim \textbf{Dirichlet}(\{\Omega_i\theta_{i\alpha}\}).$$

Using this particular method to generate the consumer uptake matrix allows us to fix its structure while including a degree of randomness. Furthermore, every uptake vector drawn from a Dirichlet distribution necessarily has a magnitude of 1, giving every consumer the same total uptake ability. If we wish to include scenarios where some consumers have a greater total uptake capacity than others, we can simply redefine the consumer uptake vectors as $\{u'_{i\alpha}\} = T_i\{u_{i\alpha}\}$, where $T_i$ is the uptake capacity of consumer $i$. Here we restrict the simulations to the case where all consumers have the same uptake capacity $T_i = 1$, the assumption being that added model complexity makes the MiCRM more difficult to approximate.

The consumer leakage tensor, which describes how consumers transform substrates into other resources, was generated analogously to the uptake matrix. Here, we must consider an additional parameter ($\phi_{i\alpha\beta}$) that encodes the probability of a substrate ($\beta$) being leaked by a given consumer ($i$) following the consumption of a particular resource ($\alpha$). Hence, for every consumer-resource pair, we define a vector $\overline{\phi}_{i\alpha}$ such that:

$$\overline{l}_{i\alpha} \sim \textbf{Dirichlet}(\overline{\phi}_{i\alpha})$$

We conducted simulations within a range of leakage magnitudes between $l^i = 0.01$ at the lower end and $l^i = 0.8$ as the most extreme case. This choice is based on empirically determined values of metabolite leakage in different strains of bacterial and co-culture experiments that suggest a feasible range of $0.05 < l^i < 0.6$ [40–43]. We also included values

above this range ($0.6 < l^i$) to explore the effects of leakage more generally. More details about our sampling procedure are provided in Section D in S1 Appendix.

## Quantifying cross-feeding and Niche overlap

Consider a community with $N$ consumers and $M$ resources; the $N \times M$ matrix $u_{i\alpha}$ describes the space of preferences consumers have for resources. The vector $\vec{u}_i$ corresponds to the $i^{th}$ row of $u_{i\alpha}$ and represents the distribution of resource preferences for the $i^{th}$ consumer.

We define Niche Overlap in the community as the average cosine similarity between each pair of consumer preference vectors $\vec{u}_i$:

$$N_o = \frac{2}{N(N-1)} \sum_{i=1}^{N} \sum_{j \neq i}^{N} \frac{\vec{u}_i \cdot \vec{u}_j}{||\vec{u}_i||||\vec{u}_j||} \tag{3}$$

This gives us a measure of how similar consumer preferences are on average, and since the consumer preference vectors are necessarily real-valued, it is a good scalar measure of competitive overlap.

On the other hand, we may ask where consumer preferences concentrate (in resource space). Averaging the sum of the preference vectors easily accomplishes this.

$$\vec{u}_{avg} = \frac{1}{N} \sum_{i=1}^{N} \vec{u}_i \tag{4}$$

Notice that $\vec{u}_{avg}$ tells us very little about how preferences vary and, indeed, nothing of how they overlap. Coupled with $N_o$, however, we can get a general notion of the shape defined by consumer preferences. While $\vec{u}_{avg}$ tells us the general direction of consumer preferences, $N_o$ is a measure of the overall spread around that average. When $N_o \approx 1$, we can surmise that consumer preferences are closely centered around $\vec{u}_{avg}$, whereas if $N_o \approx 0$, they are likely spread out.

To quantify how a given consumer is expected to leak resources in a complex community, we defined an effective leakage measure, $\vec{L}_i^{eff}$, which is simply the overall sum of resource-specific leakage vectors, weighted by the consumer's uptake capacity for the corresponding resources.

$$\vec{L}_i^{eff} = \sum_{\alpha=1}^{M} u_{i\alpha} \vec{l}_{i\alpha}$$

Where $\vec{l}_{i\alpha} = (l_{i\alpha 1}, l_{i\alpha 2}, \ldots, l_{i\alpha M})$ is the $i^{th}$ consumer's leakage vector for resource $R_\alpha$. This essentially measures the distribution of metabolites leaked by a consumer in a scenario where every resource is available at equal concentrations.

Having defined effective leakage, we can measure the extent to which resources leaked by consumers will contribute to the community's growth. To do this, we calculate the average pairwise cosine similarity between effective leakage and consumer uptake vectors.

$$C_{feed} = \frac{2}{N(N-1)} \sum_{i=1}^{N} \sum_{j \neq i}^{N} \frac{\vec{L}_i^{eff} \cdot \vec{u}_j}{||\vec{L}_i^{eff}||||\vec{u}_j||}$$

Therefore, $C_{feed}$ is an overall measure of similarity between consumer uptake preferences and the distribution of resources that are likely to be leaked by consumers. Notice that we can reframe this cross-feeding measure in terms of average effective leakage, $\vec{L}_{avg}^{eff}$.

$$\vec{L}_{avg}^{eff} = \frac{1}{N} \sum_{i=1}^{N} \vec{L}_i^{eff}$$

Hence we can consider $C_{feed}$ to be somewhat dependent on the similarity between $\vec{u}_{avg}$ and $\vec{L}_{avg}^{eff}$. More importantly, we can note that $\vec{L}_i^{eff}$ covaries with $\vec{u}_{avg}$ since they both depend on consumer uptake vectors $\vec{u}_i$.

## Stability and reactivity

To evaluate the correspondence of stability and reactivity between the MiCRM and the GLVA, we calculated their respective Jacobian matrices at steady state, $A_{Mi}$ and $A_{LV}$, as well as the Hermitian parts of $A_{Mi}$ and $A_{LV}$ - $H(A_{Mi})$ and $H(A_{LV})$ Section B in S1 Appendix. Following this, stability characterised by the leading eigenvalues of the respective pairs of Jacobian matrices $\lambda_{max}(A_{Mi})$ and $\lambda_{max}(A_{LV})$, and reactivity the leading eigenvalues of the Hermitian parts $\lambda_{max}[H(A_{Mi})]$ and $\lambda_{max}[H(A_{LV})]$.

## Quantifying the GLVA's accuracy

To quantify the GLVA's accuracy, we use the log-ratio of the GLVA- and MiCRM-predicted consumer abundances to obtain an instantaneous measure of error:

$$Err(t) = \frac{1}{N} \sum_{i=1}^{N} \log \left( \frac{C_i^{LV}(t)}{C_i^{Mi}(t)} \right)$$

Using this quantity, we define both trajectory and equilibrium abundance errors. Trajectory error is measured by integrating $Err(t)$ over the time it takes the system to equilibrate ($t_{eq}$),

$$Err_{traj} = \frac{1}{t_{eq}N} \int_0^{t_{eq}} \sum_{i=1}^{N} \log \left( \frac{C_i^{LV}(t)}{C_i^{Mi}(t)} \right) dt,$$

while equilibrium error is simply the instantaneous error rate at $t_{eq}$:

$$Err_{eq} = \frac{1}{N} \sum_{i=1}^{N} \log \left( \frac{C_i^{LV}(t_{eq})}{C_i^{Mi}(t_{eq})} \right)$$

This method of calculating accuracy has the advantage of being insensitive to relatively small discrepancies and, at the same time, adequately capturing when there are more significant disagreements between models. This can be made obvious by noting that the error measure diverges when either model predicts an extinction that does not occur in the other.

## Results

### Cross-feeding strength and Niche overlap influence GLVA accuracy

To determine how the GLVA's accuracy varies with metabolic leakage, we compared the trajectory and equilibrium error rates ($Err_{traj}$ and $Err_{eq}$) in GLVA simulations relative to the underlying dynamics in the MiCRM. Fig 2 illustrates how average $Err_{traj}$ and $Err_{eq}$ increase with the magnitude of metabolite leakage. Furthermore, at low leakage the distribution of both error rates is well behaved and localized around a small vicinity of $Err = 0$, whereas the variance of this distribution increases dramatically when $l^i \geq 0.2$.

Additionally, we find that an interaction between leakage and niche overlap (NO) plays an important role in determining the GLVA's accuracy. Specifically, Fig 2 shows how the error rate in communities with moderate and high NO is much more sensitive to the effects of leakage, displaying extreme changes in the median values and variance; in stark contrast to the modest shifts observed in communities with low NO. On the other hand, we found no clear relationship between effective leakage stoichiometry ($C_{feed}$) and GLVA accuracy.

We note that most of the deviations in the predicted GLVA abundances stem from the fact that those particular communities converge on a subset of the consumer species' richness relative to the corresponding "real" systems; that is, they

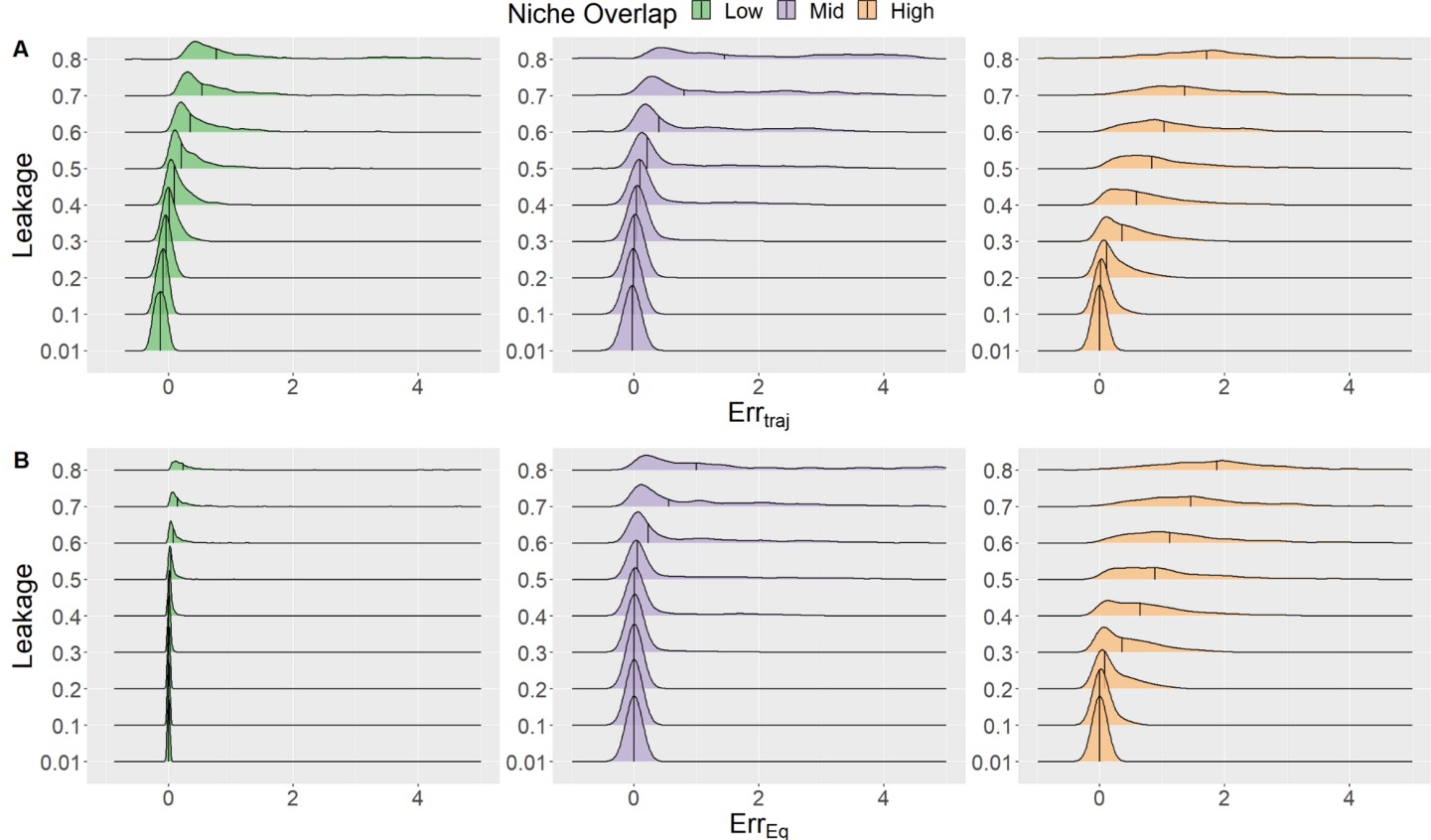

**Fig 2**. **Effect of niche overlap and metabolite leakage on the GLVA's accuracy**. Distribution of $Err_{traj}$ (**A**) and $Err_{eq}$ (**B**) for increasing magnitudes of leakage at low (green), medium (purple) and high (orange) niche overlap. The median and the spread of both error types increases with leakage and and niche overlap.

do not reach the same composition. Lastly, the biological relevance of such high leakage regimes as those considered by $l^i \geq 0.6$ is questionable, but it provides a useful limiting case.

### GLVA underestimates microbial community stability

While a simple comparison of GLVA trajectories and equilibria relative to MiCRM simulations gives a picture of the GLVA's ability to predict consumer abundances, it gives us no information on their local stability and reactivity [44]. That is, while both models may follow similar trajectories and settle on identical equilibria, this does not guarantee that they will respond to perturbations equivalently.

A direct comparison between the real parts of $\lambda_{dom}(A_{Mi})$ and $\lambda_{dom}(A_{LV})$, illustrated in Fig 3**A** reveals some important discrepancies. The GLVA's dominant eigenvalues typically have a less negative real part than the MiCRM, indicating lower stability (i.e., longer return times after perturbations), with greater leakage pushing them closer to the imaginary axis relative to the corresponding MiCRM dominant eigenvalues. The GLVA, in general, predicts that communities are less stable than their MiCRM equivalents, with the discrepancy increasing with metabolite leakage. Lastly, we can see that niche overlap has a general destabilizing effect in the MiCRM (on average, dots get bigger from left to right) but not in the GLVA; a feature which can be appreciated from the lack of a distinct size gradient in the vertical direction of Fig 3**A**.

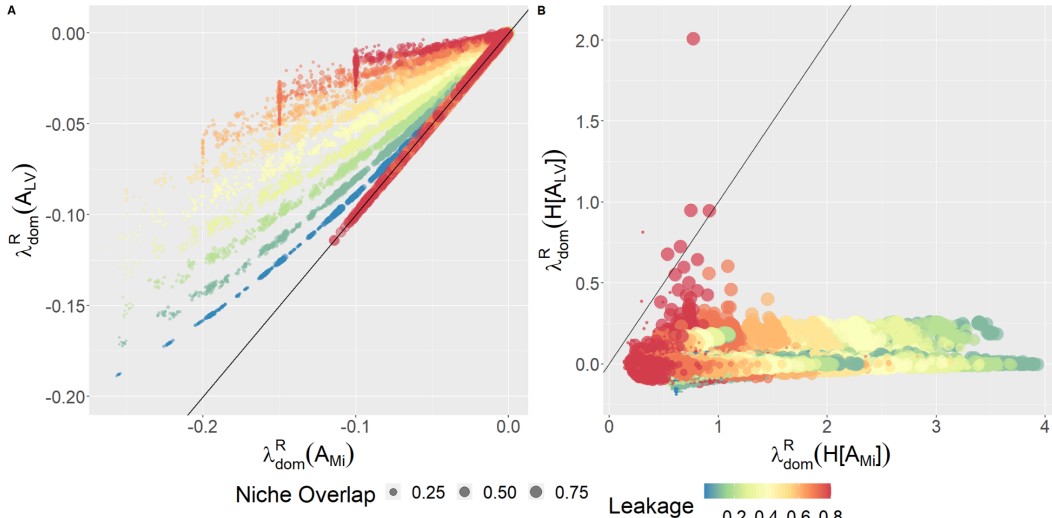

**Fig 3. The GLVA underestimates both community stability (A) and reactivity (B).** The black lines represent 1:1 correspondence. All points lie on or above the black line in A, indicating lower-than-MiCRM stability, while all the points lying below the black line in B indicate higher-than-MiCRM reactivity. Dot colors correspond to the magnitude of leaked resources in the community while dot size representing the degree of niche overlap, with dot size increasing with NO.

Turning our attention to reactivity, we observe a starker disagreement between models. Whereas fixed points in the MiCRM are universally reactive, $\text{Re}(\lambda_{dom}[H(A_{Mi})]) > 0$, the same is not true for the GLVA. In fact, more than half of the evaluated GLVA Hermitians had eigenvalues with negative real components. That is, these GLVA systems transiently dampened perturbations as opposed to the (ostensibly universal) amplification observed in the MiCRM.

## GLVA failure predicted by breakdown of timescale separation

To better understand the GLVA's inaccuracies, we need to consider two key issues: violation of the fast resource dynamics assumption and non-additivity of interactions arising from higher-order terms in the consumer-resource dynamics. Evaluating additivity directly poses a particular challenge given the combinatorial nature of higher-order interactions. For example, a second-order analysis for a community of N consumers would require quantifying $\sim N^3$ interaction coefficients between species triplets (see [45]). Even with simplifying symmetries and trivial pairs (e.g. $B_{ijk} = B_{ikj}$ and $B_{ijj} = B_{jji} = B_{jij} = 0$), the number of coefficients would grow as $N(N{-}1)(N{-}2)$. Fortunately, the time scales of resource dynamics can be quantified rather straightforwardly as a method to predict the likelihood of the GLVA's failure as we now show.

The validity of the assumption that resources reach equilibrium faster than consumers, which allows the time-scale separation for the approximation (Section A in S1 Appendix) depends upon the difference between "characteristic" time scales. More precisely, the approximation will be within $\mathcal{O}(\varepsilon)$ of the real solution for times up to $t \sim \mathcal{O}(\varepsilon^{-1})$, where $\varepsilon$ represents the relationship between the slow (consumers) and fast (resources) time scales [46,47], which can be quantified as:

$$\varepsilon = \frac{\tau_C}{\tau_R} \tag{5}$$

Here, $\tau_C$ and $\tau_R$ are the characteristic time scales of consumers and resources, respectively (further details in Section C in S1 Appendix).

To evaluate how the magnitude of $\varepsilon$ affects the GLVA's accuracy, we calculated the return times for consumers and resources directly from the MiCRM's Jacobian matrix at steady state (Section C in S1 Appendix). We then compared the return time ratios between consumers and resources and chose the smallest as a proxy for $\varepsilon$.

Fig 4A shows the relationship between the absolute equilibrium error $|Err_{eq}|$ and ($\varepsilon$). At small $\varepsilon$ simulations remain within a narrow range of low $|Err_{eq}|$, however, as the timescales of resources approach a similar magnitude to those of consumers ($\varepsilon \to 1$), we see a sharp increase in $|Err_{eq}|$. However, focusing on $\varepsilon$ alone gives us insufficient information to predict whether a particular simulation will fail. Instead, we must consider the separation of time scales in relation to the system as a whole. If the system reaches a stable equilibrium before exceeding the time frame that limits the approximation's validity ($t \sim \mathcal{O}(\varepsilon^{-1})$), we expect the approximation to remain within a small margin of error.

We can confirm this last point by comparing $\varepsilon^{-1}$ to the time to reach equilibrium in each simulation of the MiCRM. Fig 4B contrasts the distributions of absolute equilibrium error vs. $\log_{10}(\varepsilon^{-1}/t_{eq})$—the magnitude of the timescale separation relative to the time to equilibrium. This 2D density plot clearly shows that the absolute incidence of error $|Err_{eq}|$ drastically increases below the theoretical threshold of $\log_{10}(\varepsilon^{-1}/t_{eq}) = 0$. In other words, the GLVA's accuracy is only guaranteed so long as the time to equilibrium is shorter than $\varepsilon^{-1}$.

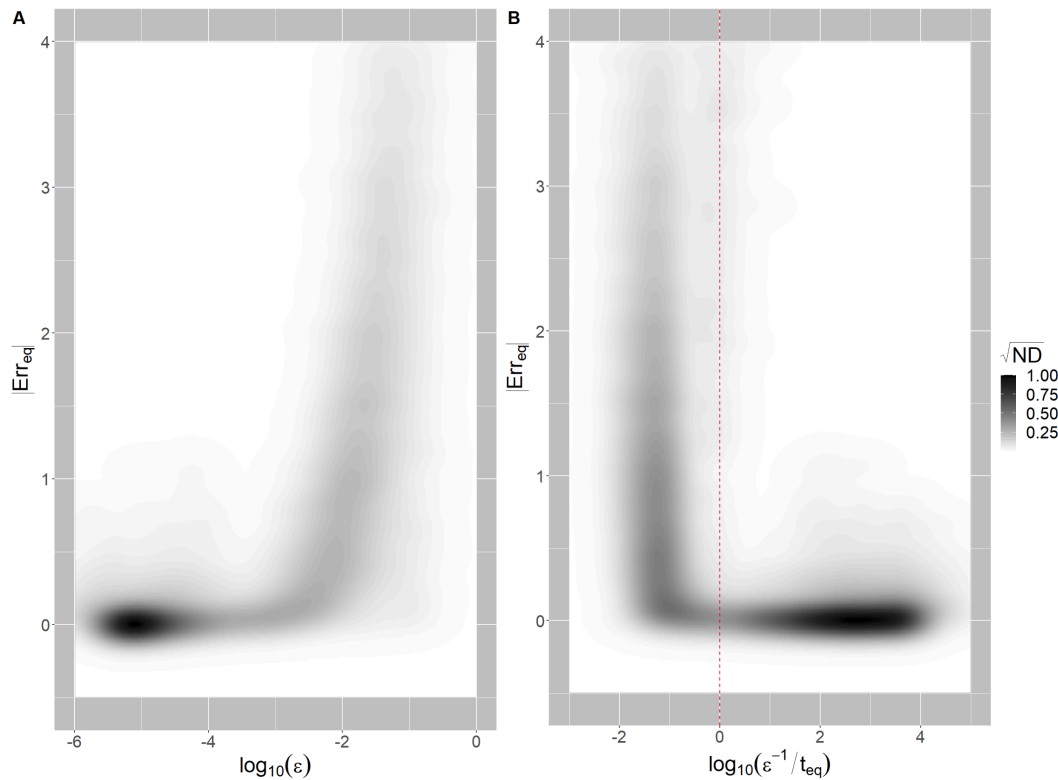

**Fig 4**. **Effect of consumer-resource timescale separation on the GLVA's accuracy**. 2D distributions of $|Err_{eq}|$ and return time ratios ($\varepsilon$) between consumers and resources in panel **A** and $log_{10}(\varepsilon^{-1}/t_{eq})$ in **B**. Darker pixels represent regions of higher probability, with the color scale darkening as the square root of the normalized density ($\sqrt{ND}$) to highlight the diffuse high error regime. The red dotted line in panel **B** corresponds to $log_{10}(x) = 0$, the theoretical lower limit of $log_{10}(\varepsilon^{-1}/t_{eq})$ above which the approximation's validity is guaranteed.

## Discussion

Although the generalized Lotka-Volterra model has been extensively studied theoretically and used empirically, its suitability for microbial communities in diverse settings remains unclear. By contrasting the dynamics of the MiCRM to its corresponding GLVA, we identified regions of parameter space where pairwise interaction terms did not accurately represent the underlying dynamics. In particular, we found that the accuracy of the GLVA declines rapidly with increasing metabolite leakage and niche overlap.

Furthermore, the GLVA's dynamics qualitatively diverge from those of the underlying MiCRM: stability is particularly sensitive to leakage, while the deviation of reactivity from the actual dynamics remains unchanged irrespective of the MiCRM's parameterization. To the best of our knowledge, the accuracy of the GLVA's dynamics in terms of reactivity has never been investigated. Our results reveal a potentially critical limitation of the GLVA, as intermittently perturbed reactive systems are more likely to settle into alternative stable states [44,48]. Hence, if the approximation undermines the MiCRM's response to perturbations, it no longer has any bearing on how the underlying system might shift under environmental or demographic stochasticity.

As mentioned in the Methods, our metric for GLVA's accuracy focuses on discrepancies in relative abundances, meaning that it is highly sensitive to differences in the predicted survival of consumer populations at steady state. This implies that the increasing incidence of errors displayed in Fig 2B is likely due to mismatches in the equilibrium community composition. Essentially, the GLVA and its corresponding MiCRM systems settle on different stable states.

Interestingly, the GLVA asymmetrically errs on the side of coexistence, as seen from the predominance of simulations where the GLVA overshoots (Fig 2, error is overwhelmingly positively skewed). Since pairwise interaction terms in the GLVA depend on resource concentrations at their steady state, this tendency to overshoot likely arises from the fact that the effective interactions differ both quantitatively (i.e., in terms of strength) and qualitatively (i.e., in terms of presence-absence or directionality) during the initial assembly stages; leading to the survival of infeasible strains before the community arrives at the interaction structure produced by the GLVA. This highlights an important point: although by construction the GLVA has an identical attractor to the MiCRM, it can settle on an alternative attractor given a different set of initial conditions, inconsistent with the actual resource-mediated consumer-consumer interactions. This makes any inferences of pairwise interactions based purely on (sufficiently frequent) temporal snapshots of abundances in co-culture highly context dependent, and limits their utility for predicting community dynamics outside of that context; a limitation that is accentuated under specific regimes (Fig 4). If, on the other hand, intra- and inter-cellular metabolite concentrations can also be tracked along with abundances, the identifiability problem that we raised in the introduction (also see [36]) is somewhat mitigated, enabling more accurate and generalisable inferences.

The GLVA's behavior at steady state differed from the MiCRM in two important respects: it systematically underestimates stability and fails to qualitatively capture the transient response to perturbations. The GLVA's tendency towards less stable interaction structures (depicted in Fig 3) reveals a key limitation of pairwise frameworks in that they disregard how resource concentrations constrain consumer populations. In effect, stability in the MiCRM directly incorporates the contributions of all resources. While the GLVA is informed by resource concentrations at equilibrium, it cannot mimic the stabilizing effect of resource limitation. This likely explains why leakage, but not niche overlap, increases the stability gap between the two models. Higher leakage leads to greater positive feedback between consumers, and in the absence of a compensatory response in resource concentrations, this has a net destabilizing effect on the GLVA.

Another detail worth noting about the stability gap is that, since stability is estimated from the dominant eigenvalues of the system at equilibrium, it is equivalent to the dynamics of the slow manifold. When leakage is small, it is virtually guaranteed that the slow manifold will depend solely on the dynamics of consumers. However, as the coupling between consumers and resources strengthens (i.e. more leakage), it increases the likelihood that resources will contribute to the dominant eigenvalues.

While the MiCRM was reactive for every community simulated, the GLVA erroneously predicted the direction of the initial response to perturbations in over half of our data points. In cases where the GLVA displayed positive reactivity, it typically underestimated the strength of the effect by an order of magnitude. Although reactivity is not usually given as much attention as stability, it has been argued that it is at least as important for understanding the effects of chronically disturbed communities [44,49].

As shown in Fig 4, the accuracy of the GLVA can be predicted from an approximate measure of the time-scale separation between consumers and resources. This apparent relationship between the temporal coupling of the consumer-resource system and the GLVA's performance suggests that the degree of separation, as measured here, serves as a good indicator of when a particular consumer-resource system can be described with consumer abundances alone. More importantly, the degree of separation defines a neighborhood around equilibrium, within which the GLVA will remain below a bounded margin of error compared to the true dynamics. This neighborhood can be understood as the set of initial conditions (likewise, perturbations) for which $t_{eq} \leq \varepsilon^{-1}$.

It should be noted that this issue was originally raised, albeit indirectly, by MacArthur [38] in terms of the symmetry of the matrix of interactions. If we take a Consumer-Resource model and assume that resources equilibrate quickly compared to the dynamics of the consumer, then the effective interaction matrix is necessarily symmetric. This result was later shown to be incorrect under a wider range of conditions. As Marsland et al. [34] have shown, the requirement of symmetric interactions can be relaxed through specific rescaling procedures. More generally, it appears that some variants of the Consumer-Resource equations are inherently asymmetric and cannot be straightforwardly rescaled, rendering the symmetry of the interaction matrix an inadequate indicator of the validity of the GLV approximation.

Although our analysis focuses on the ability of GLVA to represent the dynamics emerging from metabolite competition and cross-feeding dynamics in the MiCRM, the results can be straightforwardly generalized to any interaction mechanism that is indirectly mediated, such as through chemical stressors [50]. Our method of estimating $\varepsilon$ (Section C in S1 Appendix) is directly applicable to any system, provided the ODEs can be defined, and the Jacobian is diagonally dominant.

As such, although the MiCRM's structure is fairly general in its ability to capture the key elements of microbial consumer-resource dynamics, the question arises whether our results apply to variants of the MiCRM (e.g., different functional responses or the inclusion of essential nutrients). In particular, the issue of timescale separation and strength of consumer-resource coupling is a universal feature of coupled dynamical systems; hence, we do not expect any meaningful differences for a reasonably modified set of ODEs. Whether these insights can be extended to accommodate more complex interaction mechanisms, such as allelopathy or chemically mediated signalling, remains to be seen.

Some systems inherently have a greater timescale separation between consumer and resource pools. For instance, metabolic scaling laws dictate that because phytoplankton are generally much larger than bacteria, they operate on slower timescales. As a result, they interact with their metabolites more slowly than bacteria do, suggesting that the GLVA might, in general, be a more accurate model for quantifying phytoplankton community dynamics.

Overall, these results provide guidelines for the appropriate use of pairwise models in the study of microbial communities. Firstly, theoreticians seeking to apply the generalized Lotka-Volterra model to relate community structure to the system's dynamic behaviors should verify that the system in question does not exhibit environmental (resource) feedbacks strong enough to invalidate the model's assumptions. This point becomes especially important when considering the impacts of environmental perturbations (e.g., temperature fluctuations [23,32,50–52]), a topic of increasing importance as we seek to quantify the effects of anthropogenic change. We show that GLV models can systematically mis-predict equilibrium resilience, often underestimating recovery rates and sometimes even predicting no transient amplification when the mechanistic system exhibits strong reactivity. This distinction is crucial for assessing robustness and the risk of regime shifts [44,48]. For empiricists, relying on GLV fits alone could mask early warning signs of collapse or invasion, since resource-mediated overshoot dynamics are not captured by static interaction terms. Validating GLV

predictions against actual perturbation experiments, rather than only steady-state outcomes, is therefore crucial to ensure resilience is represented correctly.

Because ($\varepsilon$) can be estimated from basic measurements of nutrient depletion, byproduct accumulation, and microbial growth rates, it offers an empirically accessible "litmus test". This complements the broader caution that GLV model fits are non-identifiable and may overfit sparse time series: multiple parameter sets can reproduce similar trajectories but differ mechanistically. Collecting additional data on metabolite or resource concentrations and perturbation responses to constrain GLV model fitting may therefore be necessary when consumer-resource coupling is strong [53–55].

Empirically determining $\varepsilon$ in microbial communities requires precise monitoring of cell densities as well as substrate and metabolite concentrations. Given well-resolved data, $\varepsilon$ can be estimated by perturbing the community and monitoring how long it takes the microbial consumer populations and substrate concentrations to return to a steady state. Another method would be to carry out single-strain culture experiments to determine each of the parameters (in this case: uptake, yield, and metabolite leakage), and use those data to calculate, using resource-explicit models like the MiCRM, the approximate timescale separation *a priori*. The robustness of the separation of timescales can be more feasibly (but less accurately) assessed from the equilibrium properties of the system, such as the proportion of resources contained in consumer biomass relative to the environment in addition to estimates of average metabolite leakage, given their role in determining characteristic timescales (see equations in Section C in S1 Appendix).

Finally, we note that our results should be interpreted with the understanding that the MiCRM assumes well-mixed conditions, linear uptake kinetics, and constant leakage fractions per species. Real microbes might exhibit more complex dynamics (e.g., threshold-driven secretion, spatial metabolite gradients, or adaptive responses) that could introduce qualitatively new higher-order effects beyond those captured by the basic MiCRM. Our findings will likely still apply; any such complexity would further undermine the accuracy of a GLVA, but the precise thresholds of timescale separation would differ.

## Conclusion

In conclusion, the magnitude of metabolite leakage and the resultant cross-feeding, mediated by niche overlap structure, has a strong bearing on the extent to which the GLVA can accurately reproduce the underlying MiCRM dynamics. In particular, we found near-zero correspondence between models regarding reactivity, highlighting the importance of higher-order interactions and the GLVA's inability to account for them in predicting perturbative behavior. We show that these inaccuracies stem primarily from (the lack of) timescale separation between consumer and resource dynamics, and provide a measure to quantify the strength of this coupling.

## Supporting information

**Section A in S1 Appendix. The generalised Lotka-Volterra approximation.** Step-by-step derivation of the GLVA from the MiCRM.

**Section B in S1 Appendix. Stability analysis.** Linearization and stability analysis for the MiCRM.

**Section C in S1 Appendix. Quantifying separation of timescales.** Motivating the importance of timescale separation and deriving an approximate measure of consumer-resource coupling.

**Section D in S1 Appendix. Balanced sampling of competitive and cooperative communities.** Details on the method for balanced sampling of communities with varying degrees of competition and cooperation.

(PDF)

## Acknowledgments

We thank Alberto Pascual-García and Tom Clegg for insightful comments on an earlier version of this manuscript.

## Author contributions

**Conceptualization:** Michael P. Mustri, Samraat Pawar.

**Data curation:** Michael P. Mustri, Quqiming Duan.

**Formal analysis:** Michael P. Mustri, Quqiming Duan.

**Investigation:** Michael P. Mustri.

**Methodology:** Michael P. Mustri, Quqiming Duan, Samraat Pawar.

**Project administration:** Michael P. Mustri, Samraat Pawar.

**Resources:** Samraat Pawar.

**Software:** Michael P. Mustri, Quqiming Duan.

**Supervision:** Samraat Pawar.

**Visualization:** Michael P. Mustri.

**Writing – original draft:** Michael P. Mustri, Samraat Pawar.

**Writing – review & editing:** Michael P. Mustri, Quqiming Duan, Samraat Pawar.

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
