## [Decision Letter · Decision Letter 0]

10 Jul 2025

PCOMPBIOL-D-25-00609

Accuracy of the Lotka-Volterra Model fails in strongly coupled

microbial consumer-resource systems

PLOS Computational Biology

Dear Dr. Mustri,

Thank you for submitting your manuscript to PLOS Computational Biology. After careful consideration, we feel that it has merit but does not fully meet PLOS Computational Biology's publication criteria as it currently stands. Therefore, we invite you to submit a revised version of the manuscript that addresses the points raised during the review process.

Please submit your revised manuscript within 60 days Sep 09 2025 11:59PM. If you will need more time than this to complete your revisions, please reply to this message or contact the journal office at ploscompbiol@plos.org. Please include the following items when submitting your revised manuscript:

We look forward to receiving your revised manuscript.

Kind regards,

Eric C. Dykeman, Ph.D.

Academic Editor

PLOS Computational Biology

Zhaolei Zhang

Section Editor

PLOS Computational Biology

**Journal Requirements:**

At this stage, the following Authors/Authors require contributions: Michael Phillip Mustri, Quqiming Duan, and Samraat Pawar. Please ensure that the full contributions of each author are acknowledged in the "Add/Edit/Remove Authors" section of our submission form.

**Reviewers' comments:**

Reviewer's Responses to Questions

**Comments to the Authors:**

Reviewer #1: In this paper the authors compare the dynamic behavior of the Generalized Lotka-Volterra (GLV) model against a microbial consumer-resource dynamics (MiCRM) taken as benchmark and quantify the concordance between them across a range of biologically different scenarios. The main result reveals that the probability of the GLV model's predictions being inaccurate can be measured by a simple, empirically accessible metric: the timescale separation between consumers and resources.

I find interesting the analysis of the relative effects of metabolite leakage and niche overlap on the GLVA’s accuracy.

The agreement of GLV to MiCRM is very good provided the leakage is low (quite independently of the degree of niche overlap), and relatively good up to medium leakage, provided the niche overlap is not high. In other words, the effect of metabolite leakage on the GLVA’s accuracy is more important than the one of niche overlap.

First of all a general comment: What about contrasting either GLV or MiCRM equations against experimental data?

As shown in Fig. 1, the MiCRM generally produces an initial overshooting of population abundances that is absent in the GLVA. A significant portion of the trajectory error arises from this discrepancy compared to the GLVA. Is such initial overshooting actually observed in experiments with microbial communities?

Comments by sections:

Results

In Fig. 3, the color codes for some of the leakage values are almost invisible. Please correct this.

Discussion

L. 372: Regarding the importance of higher-order interactions to overcome the GLVA failures: A model's failure might actually stem from a misspecified pairwise interaction, rather than the presence of higher-order interactions, as pointed out by Case & Bender (1981).

Appendix A

A question about Eq. (15), which defines the interaction matrix coefficients a_ij as a sum of products. Specifically, of u_i\alpha times the proportion of resources leaked times the partial derivative of R_a respect to C_j. Given that the first two factors are, by definition, non-negative, it appears that the sign of α_ij is primarily determined by the sign of the partial derivative ∂C_j/∂R_a. Is this interpretation correct? How do you interpret this dependency? Do you have an estimate of the relative prevalence or weights of competitive (negative) versus facilitative (positive) interactions predicted by this formulation?

Minor comments:

- L. 105: MacArthur appears twice.

- In the equation of the error trajectory, above L. 183, a time differential, dt, is lacking in the integral.

Reviewer #2: The manuscript by Mustri et al. attempts to assess how well a Lotka-Volterra model (that assumes direct interactions) approximates population dynamics arising from a model that assumes resource-mediated interactions. To derive the Lotka-Volterra approximation, they take resource dynamics to quasi steady state and then linearize the resulting interaction terms around the consumer equilibrium (I believe that this type of analysis of consumer-resource models goes back to MacArthur in the 70s.). They then compare dynamics and measures of stability and reactivity of the resulting approximate Lotka-Volterra system to those of the resource-mediated interaction model.

1) Throughout, implicitly the MiCRM was taken to be “truth” and the GLVA was compared to this truth. It should be noted that the GLVA as derived is not the same as what would be inferred by using a statistical method to fit the time series data generated by a MiCRM. For example, the best fit GLV model to the MiCRM time series in figure 1 might look far more like the MiCRM time series. Also, it would have different stability and reactivity than the GLVA. It is not at all clear from the analyses in this manuscript that an inferred GLV model would actually differ in (for example) estimated stability and reactivity in the same way the GLVA differs.

2) Throughout, error was defined by how often and by how much the population dynamics deviate rather than a comparison of the nature and relative strengths of the interactions themselves. Often the goal of fitting a GLV model to data is not to make predictions about future dynamics, but rather to study the interactions themselves. Even if the dynamics differ somewhat, are the nature of interactions accurately estimated (for example negative interactions from resource competition or positive interactions from cross feeding)? To me, this would be a more interesting test of the accuracy of the GLV even if dynamics differ subtly.

3) The GLV approximation is performed by taking the resources to quasi steady state and then linearizing the per capita rates about the equilibrium. Mathematically the QSS assumption is identical to setting dR/dt=0 as stated in the text in the Appendix. But what is actually being assumed is not that R is at equilibrium and not changing, but rather that it changes very rapidly so that there is a small parameter epsilon such that epsilon*dR/dt=f(R,C), and then epsilon is set to zero. Intuitively, I believe that this requires that nearly all biomass is tied up in consumers C rather than as free biomass R (large uptake rates relative to the decay rates m and relatively low leakage). Essentially, free resource would be nearly instantly taken up by consumers if the QSS assumption is valid. It seems to me that leakage increases the stability gap because it makes the QSS assumption invalid (resources would no longer be fast relative to consumers). Working this through formally may show exactly which combinations of parameters need to be small for the approximation to be good. The proportion of free resource at steady state would also be measurable, so if my intuition above is correct it would be something that could be ascertained relatively easily in real systems.

4) I don't think that the linearization for the approximation of the GLV line 381-384 need not be around the equilibria for C in order to create a GLV model. This is of course a natural choice because it ensures that the GLV will have the same equilibria as the consumer resource model, but we wouldn’t expect the approximation to work well as C deviates from its equilibrium. Other choices might make sense in other scenarios, such as wanting to predict the initial growth rates of consumers when given a suite of resources (community assembly).

5) The MCRM also makes the assumption that growth rates of consumers due to different resources are additive (just like interactions in the GLV). Namely, if a consumer grows at rate r1 on resource 1 and rate r2 on resource 2, the MCRM model assumes that the growth rate of the consumer will be r1+r2. Yet we know this generally not the case. The consumer may still just grow at rate r1 even when resource 2 is present because it prefers growth on resource 1 over on resource 2.

6) Line 189: Why were the results classified as accurate, overshot and undershot rather than simply showing the relative error directly? Fig 2: I’d have rather seen a plot of relative error as a function of leakage. Also, I would not consider deviations in predicted dynamics of 20% as large. If we could actually predict even the order of magnitude of dynamics in real systems we would probably be happy.

7) How do the chosen leakage levels correspond to real systems? Low leakage is defined as 0.3 but how is this number selected? Why not 0.1 or 0.01? If leakage is typically low in real systems, maybe the GLV provides an adequate approximation. More generally, the results depend heavily on the specific numbers chosen and there is little justification for these from real systems.

Minor issues:

Figure 1: The vertical axes in both plots should be the same for ease comparison (alternatively these could be shown on the same plot).

Figure 3: It is very difficult to see the definition of the colors

Reviewer #3: The manuscript examines the conditions under which a microbial community represented by a consumer-resource model can be adequately captured (in terms of its steady state composition) by a generalized Lotka-Volterra approximation. Even though the ideas in the paper are not completely novel, the manuscript offers a comprehensive view of the relevant concepts and parameters related to the consumer-resource model. The biological insights offered in the manuscript are, in my opinion, helpful for the community and I have an overall favorable view of this manuscript.

Major comments:

1. Line 207: While the difference in predicted stability is shown clearly, there is not enough discussion on the reason behind this difference in my opinion. I would suggest including the reason (or your speculation) about this difference in stability between the two models.

Minor comments:

1. Line 193: I suggest using a more explicit and direct title for this result section. Using ‘community structure’ is a little vague. Perhaps mentioning niche overlap would be a more direct representation of the concepts discussed in this section.

2. Figure 3: Please include more information about each panel in the caption to make the figure more accessible.

3. Figure 3: Please make the colors in the legend more prominent (larger dots?)

4. In Appendix C, the discussion about the harmonic oscillator is distracting, in my opinion. I think it would be better to focus on the consumer-resource model instead. If an example is needed, I would suggest discussing a simple two-species community in the consumer-resource model for more insight.

5. Line 224: The title of this section is a little vague. Please consider replacing it with a more direct and explicit title that describes your findings.

6. The discussion section does a good job of summarizing the findings and linking them to the existing literature, but it’s relatively weak in discussing limitations and future directions.

**Have the authors made all data and (if applicable) computational code underlying the findings in their manuscript fully available?**

Reviewer #1: None

Reviewer #2: Yes

Reviewer #3: Yes

PLOS authors have the option to publish the peer review history of their article (what does this mean?). If published, this will include your full peer review and any attached files.

Reviewer #1: No

Reviewer #2: No

Reviewer #3: No

**Figure resubmission:**
---

## [Decision Letter · Decision Letter 1]

7 Nov 2025

Dear Mr. Mustri,

We are pleased to inform you that your manuscript 'Accuracy of the Lotka-Volterra Model fails in strongly coupled

microbial consumer-resource systems' has been provisionally accepted for publication in PLOS Computational Biology.

Best regards,

Eric C. Dykeman, Ph.D.

Academic Editor

PLOS Computational Biology

Zhaolei Zhang

Section Editor

PLOS Computational Biology

Reviewer's Responses to Questions

**Comments to the Authors:**

Reviewer #1: ALL COMMENTS HAVE BEEN ADRESSED

Reviewer #3: In my opinion, the authors have adequately addressed the comments raised by reviewers.

**Have the authors made all data and (if applicable) computational code underlying the findings in their manuscript fully available?**

Reviewer #1: Yes

Reviewer #3: Yes

PLOS authors have the option to publish the peer review history of their article (what does this mean?). If published, this will include your full peer review and any attached files.

Reviewer #1: No

Reviewer #3: No

---

## [Editor Report · Acceptance letter]

PCOMPBIOL-D-25-00609R1

Accuracy of the Lotka-Volterra Model fails in strongly coupled

microbial consumer-resource systems

Dear Dr Mustri,

I am pleased to inform you that your manuscript has been formally accepted for publication in PLOS Computational Biology. Your manuscript is now with our production department and you will be notified of the publication date in due course.

With kind regards,

Anita Estes
